**Data Availability Statement:** All relevant data are within the paper and its Supporting Information files.

# Variation in the serotonin transporter genotype is associated with maternal restraint and rejection of infants: A nonhuman primate (*Macaca mulatta*) model

Elizabeth K. Wood[1¤a]*, Zachary Baron[2], Ryno Kruger[1¤b], Colt Halter[1¤c], Natalia Gabrielle[1¤d], Leslie Neville[2], Ellie Smith[1], Leah Marett[2], Miranda Johnson[1], Laura Del Rosso[3], John P. Capitanio[3,4], J. Dee Higley[1,2]

1 Department of Psychology, Brigham Young University, Provo, Utah, United States of America,
2 Department of Neuroscience, Brigham Young University, Provo, Utah, United States of America,
3 California National Primate Research Center (CNPRC), Davis, California, United States of America,
4 Department of Psychology, University of California—Davis, Davis, California, United States of America

¤a Current address: Department of Psychiatry, Oregon Health & Science University, Portland, Oregon, United States of America
¤b Current address: Department of Psychology, Emory University, Atlanta, Georgia, United States of America
¤c Current address: Department of Psychology, Wayne State University, Detroit, Michigan, United States of America
¤d Current address: Department of Psychology, Utah Valley University, Orem, Utah, United States of America
* woodel@ohsu.edu

## Abstract

Studies show that maternal behaviors are mediated by the bivariate serotonin transporter (*5-HTT*) genotype, although the findings are mixed, with some studies showing that mothers with the *s* allele exhibit increased maternal sensitivity, while other studies show that mothers with the *s* allele show decreased maternal sensitivity. Nonhuman primate studies offer increased control over extraneous variables and may contribute to a better understanding of the effects of the *5-HTT* genotype on maternal sensitivity. This study assesses the influence of *5-HTT* genotype variation on maternal sensitivity in parenting in 125 rhesus macaque mothers (*Macaca mulatta*) during the first three-months of their infants' lives, an age well before typical infants undergo weaning. Mothers were genotyped for the *5-HTT* genotype and maternal behaviors were collected, including neglectfulness, sensitivity, and premature rejections during undisturbed social interactions. Results showed that mothers homozygous for the *s* allele rejected their infants the most and restrained their infants the least, an indication that mothers with the *s* allele are more likely to neglect their infants' psychological and physical needs. These findings suggest that, at an age when an infant's needs are based on warmth, security, and protection, mothers with an *s* allele exhibit less sensitive maternal behaviors. High rates of rejections and low rates of restraints are behaviors that typically characterize premature weaning and are inappropriate for their infant's young age. This study is an important step in understanding the etiology of variability in maternal warmth and care, and further suggests that maternal *5-HTT* genotype should be examined in studies

**Funding:** This work was supported by:
R24OD010962 (JPC) and P51OD011157 (CNPRC
base grant), as well as small mentoring grants
from Brigham Young University. The funders had
no role in study design, data collection and
analysis, decision to publish, or preparation of the
manuscript.

**Competing interests:** The authors have declared
that no competing interests exist.

assessing genetic influences on variation in maternal sensitivity, and ultimately, mother-infant attachment quality.

## Introduction

Differences in mother-infant attachment quality are widely believed to be rooted in the early interactions between caregivers and their infants, principally based on maternal sensitivity to an infant's physical, emotional, and temperamental needs. A mother's ability to modify her response according to her infant's needs is a critical factor in determining mother-infant attachment quality and socioemotional outcomes [1, 2]. Some mothers, such as those who are depressed, are less likely to modify their behaviors according to their infants' age-related needs, resulting in infants ultimately developing an insecure attachment [3]. In more typical populations, mothers with high sensitivity to infant physical and emotional needs are more likely to promote secure mother-infant attachments [4, 5]. Maternal sensitivity to their infants' changing developmental needs is also associated with infant development of social competence, greater impulse control, fewer tantrums, and less negative emotionality, when compared to mothers who fail to modify their behaviors to match their infants' emotional state [6, 7].

Many factors can impact maternal sensitivity, including a mother's own early life experiences [8], socio-economic status [9], mental health [10, 11], and the quality of her social support [12]. Some studies suggest that maternal genetic factors may also play an important role in maternal sensitivity [13], although such genetic studies of maternal sensitivity are infrequent. The effect of maternal serotonin transporter (*5-HTT*) genotype on maternal sensitivity has received a good deal of attention (see Bakermans-Kranenburg [14] and Landoni et al. [15] for reviews and meta-analyses). However, these studies show somewhat inconsistent findings, with some showing that mothers who possess the short (*s*) allele exhibit *more* sensitivity toward their infants [16, 17], whereas other studies show that mothers possessing the *s* allele exhibit *less* maternal sensitivity [13, 18–20]. Mileva-Seitz et al. [16] posit that this discrepancy may be related to differences in the populations sampled or differences in the assessment environments (laboratory vs. home setting), an interpretation that is corroborated by research showing that there are differences in mothering behavior when behaviors in the home and laboratory settings are compared [21]. Bakermans-Kranenburg & van Ijzendoorn [14] suggest that a mother with an *s* allele may be attentive to her child's emotional signals, and therefore respond promptly and sensitively to their child when compared to a mother with the long (*L*) allele. On the other hand, in response to a difficult child, they may become overwhelmed, an interpretation reminiscent of Thomas and Chess' concept of goodness-of-fit of temperament between parents and their offspring [22].

Mileva-Seitz [16] suggest that the mixed findings may be a result of varied environmental backgrounds (i.e., gene-by-environment interactions). For example, Sturge-Apple et al. [23] showed that mothers that possess an *s* allele are especially sensitive to environmental context, finding that mothers with the *s* allele exhibited insensitive maternal behavior and were more likely to use harsh parenting, but only when the environment was stressful (as measured by a high degree of inter-parental conflict). Sawano et al. [24] found that maternal *5-HTT* genotype led Japanese mothers to express negative affect toward their infant, but only when the mothers had experienced poor early maternal care themselves. Similarly, Morgan et al. [20] found a positive association between disruptive child behavior and parental negativity, and this relationship was stronger among parents who possessed an *s* allele. In a study of maternal sensitivity, Baiao et al. [25] found that mothers possessing the *s* allele were especially sensitive to

environmental context, with mothers that were homozygous for the *s* allele exhibiting the highest and lowest level of maternal sensitivity depending on high or low family support, respectively. While mothers with the *L* allele showed a similar response to high and low family support, the influence of the environment was attenuated.

In an effort to reconcile these discrepancies, to increase overall understanding of the effect of *5-HTT* genotype on maternal sensitivity, and to overcome the difficulties inherent in human research, this study utilizes a nonhuman primate model. Rhesus monkeys (*Macaca mulatta*) are ideally suited to assess the relationship between genetic influences and maternal behavior. There is a long and rich scientific history of using rhesus monkeys to model the human mother-infant bond [26]. Much of what is understood about human mother-infant attachment behavior comes from studies of nonhuman primates [27, 28]. They are closely related to humans, sharing about 93% of their genetic sequence [29], including an orthologous biallelic *5-HTT* genotype [30]. Like humans, rhesus monkeys also exhibit extended development [31]. Moreover, their environments can be closely controlled, reducing uncontrolled variance, and allowing for the detection of small genetic effects. Paralleling human development, rhesus monkey infants are born highly altricial, spending nearly 90% of their time on their mother's ventrum during the first month of life [32]. Like humans, at this early age they lack the sophistication to recognize inherent risks in the environment. Consequently, as they begin to leave their mother's side, mothers protect their infants by restraining and retrieving them as they attempt to explore their environments and interact with peers, typically showing a high degree of protection through the third month of life. Maternal rejections before this age are rare and are considered atypical [33–37], and result in pathological behaviors in the infant, such as anxiety-like behaviors and aggressiveness [36–38]. When rhesus monkey infants reach about three months of age, the weaning process begins, reaching its peak by six months of age, about the same time the breeding season begins, and typical mothers exhibit infant rejections at this more advanced infant age [37, 39–41]. Using a rhesus macaque model, this study investigates the influence of the *5-HTT* genotype on maternal sensitivity. Specifically, it is hypothesized that, at an age when infants are still dependent on their mother to act as a secure base and to protect them, mothers possessing the *s* allele of the *5-HTT* genotype will exhibit inappropriate parenting behaviors, such as premature maternal rejections of their immature, dependent infants. As a corollary, it is hypothesized that rhesus mothers with the *s* allele will be less likely to keep their infant in close proximity, as measured by restraining and retrieving their infants during attempts at exploration.

## Methods

Subjects were $N = 125$ rhesus macaque (*Macaca mulatta*) mother-infant dyads housed at the California National Primate Research Center (CNPRC) in Davis, California. All infants were born and reared in large outdoor, half-acre corrals comprised of 50-to-125 subjects, living in social conditions that closely-approximate the natural, species-specific social composition found in the wild [42]. Infants were approximately three months old ($M_{age} = 3.10 \pm 0.08$ months) and mothers were approximately seven years old ($M_{age} = 7.16 \pm 0.28$ years) at the time of the study. With the exception of one subject that was reared in the nursery before introduction into the larger, outdoor social group into which her infant was born, the early life history of all of the subjects was identical, including being born into and reared in the large, outdoor corrals. All data were collected between the years of 2016–2019 and all research was conducted in compliance with protocols established by the University of California at Davis' Animal Care and Use Committee.

## Behavioral observations

All mother-infant dyads were observed for four, 300-second sessions by trained observers, with an inter-rater reliability of $r > 0.85$ or above. Observers were blind to the objectives of the study and were naïve with regard to the genotypes of the observed subjects. Maternal behaviors were recorded as modified frequencies per 300 seconds, using an established mutually exclusive, exhaustive ethogram (see Table 1). Briefly, recorded behaviors included maternal contact with infant, including mutual-ventral contact, maternal grooming of infant, and maternal restraint, rejection, and retrieval of infant. To account for potential behavioral variability due to time of day (morning vs afternoon), the four observations were randomly distributed, with two in the morning and two in the afternoon. Individual differences in maternal behaviors tend to be consistent over time, even after accounting for offspring age [43, 44]. To assess whether the maternal behaviors were interindividually stable and consistent for each of the measured behaviors, a correlation between the average of the first two behavioral observations and the second two behavioral observations was conducted. Results indicated that the behaviors were positively correlated and statistically significant ($p < .05$), an indication of stable, reliable behavioral assessments. Rushton et al. [45] show that by aggregating across observations when behaviors are positively correlated, power is increased. Thus, all analyses used the overall mean of each behavior. Observations were conducted in the summer months at the tail-end of the birthing season—mid-June through the end of August [46].

## Genotyping

Blood was obtained from the mothers as part of a larger research program at the CNPRC [47]. DNA was extracted using a standard phenol-chloroform protocol. Genotyping for *5-HTT* was done by fractionating PCR products in 2% agarose gels and by capillary electrophoresis in ABI 3730 DNA Analyzer (Applied Biosystems, Foster City, CA, USA) [for a detailed description of the methodology, see 48]. The distributions of genotypes were as follows: $n = 50$ mothers were homozygous for the *L* allele, $n = 58$ mothers were heterozygotes, and $n = 14$ mothers were homozygous for the *s* allele. Three subjects possessed rare *5-HTT* variants (i.e., *XL, XXL*) and they were excluded from the analyses. Genotype frequencies did not significantly deviate from Hardy-Weinberg equilibrium ($\chi^2 = 0.21$, df = 2, $p = .90$).

## Data analysis

Preliminary analyses suggested that there were no effects of infant sex, infant age, maternal parity, mother's age, or observation year on outcome variables of interest ($p > .05$), thus, these variables were excluded from formal analyses. To assess the relationship between maternal

**Table 1. Ethogram for maternal behavior.**

| Behavior | Definition |
|---|---|
| Contact | Mother is in physical contact with her infant, but not in mutual-ventral contact |
| Groom | Mother grooms her infant (fingers or mouth) |
| Mutual-ventral Contact | The ventrums of the mother and her infant are touching |
| Reject | Mother refuses infant's attempts to approach or make contact by blocking, pushing, or pulling her infant away from her or terminating contact |
| Restrain | Mother grabs, holds, or tugs at her infant's attempts to leave proximity |
| Retrieve | Mother approaches her infant and makes physical contact |

All behaviors were recorded using modified frequency sampling during four, 300-second observation periods.

parenting behaviors and *5-HTT* genotype, separate between-groups ANOVAs were conducted with maternal *5-HTT* genotype as the independent variable and the mean frequency of each maternal behavior as the dependent variable. Differences between groups were assessed using *a priori* planned comparisons. To account for multiple comparisons, an overall MANOVA was also conducted, with maternal *5-HTT* genotype as the independent variable and the maternal parenting behaviors as the dependent variables. As noted above, while the majority of subjects were born and reared in the larger corrals, one mother (heterozygous for the *5-HTT* genotype) was reared in the nursery. Preliminary analyses assessing the impact of including this subject showed no difference in the direction or significance of effects, so they were included in all final models. All analyses were conducted in SPSS, version 26 (IBM, 2019), with alpha set at $p < .05$.

## Results

Results indicated a significant effect of maternal *5-HTT* genotype on the frequency of maternal rejections ($F(2,117) = 3.71$, $p = .03$), with mothers that were homozygous for the *s* allele rejecting their infants more often, on average, when compared to mothers that were homozygous for the *L* allele ($p < .03$). Mothers that were homozygous for the *L* allele also exhibited fewer rejections, when compared to heterozygous mothers ($p < .03$; *LL*: $M = 0.03 \pm 0.07$; *Ls*: $M = 0.11 \pm 0.21$; *ss*: $M = 0.15 \pm 0.35$; see Fig 1 and S1 Fig).

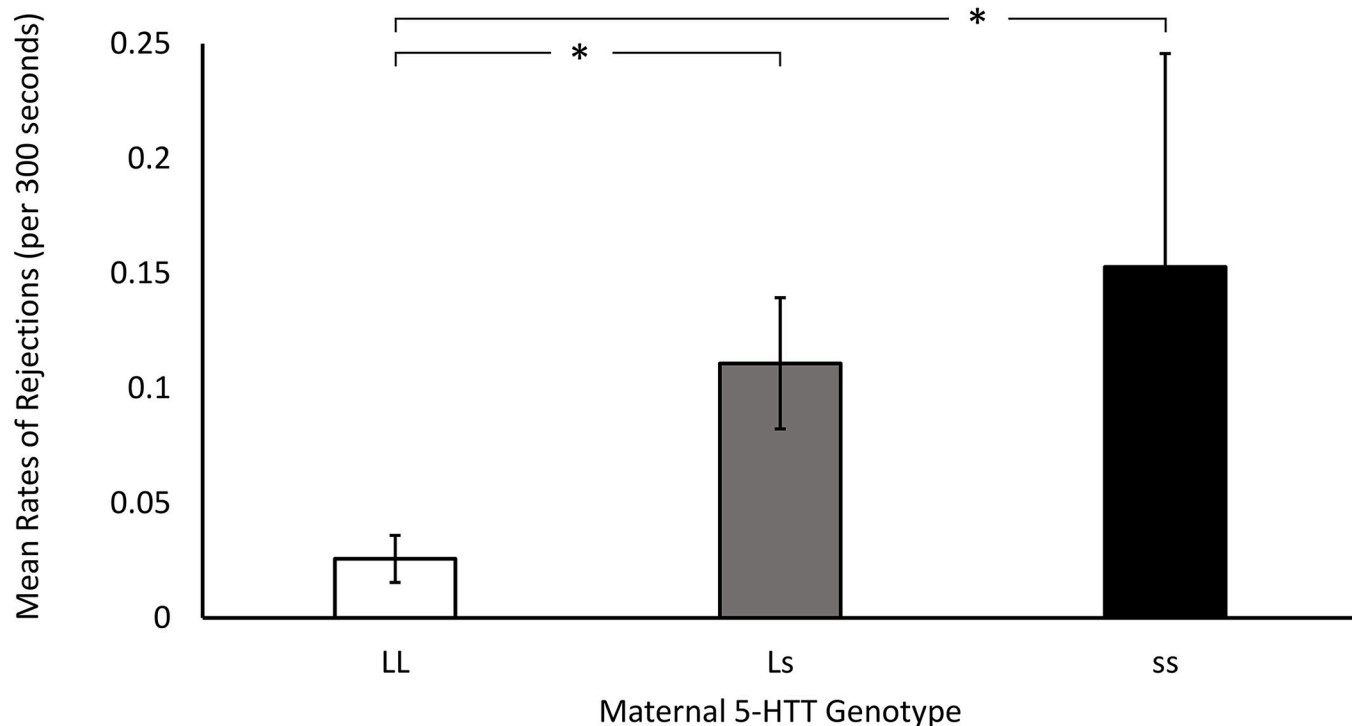

**Fig 1. Effects of maternal *5-HTT* genotype on the frequency of maternal rejections of infants.** When compared to mothers that were homozygous for the *L* allele, mothers that were homozygous for the *s* allele exhibited higher rates of infant rejections ($p = .03$). Mothers that were homozygous for the *L* allele also exhibited fewer rejections, when compared to mothers that were heterozygous ($p = .03$). White bars indicate mothers that were homozygous for the *L* allele, gray bars indicate heterozygous mothers, and black bars indicate mothers that were homozygous for the *s* allele. Error bars are standard errors.

## Effects of Maternal 5-HTT Genotype on the Frequency of Maternal Restraints of Infants

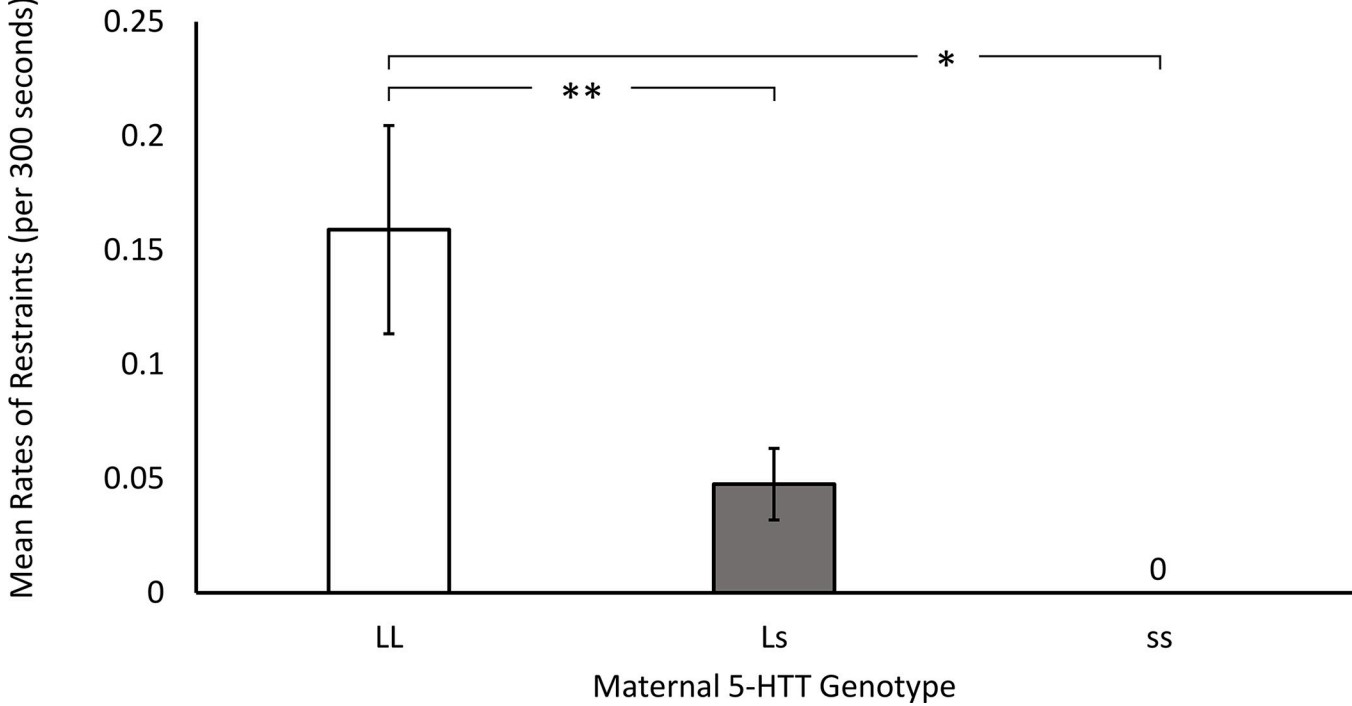

**Fig 2. Effects of maternal *5-HTT* genotype on the frequency of maternal restraints of infants.** When compared to heterozygous mothers and mothers that were homozygous for the *s* allele, mothers homozygous for the *L* allele exhibited the highest rates of maternal restraints (*p* = .02). Mothers that were homozygous for the *L* allele restraining their infants more, on average (*p* < .01), when compared to heterozygous mothers and mothers homozygous for the *s* allele (*p* < .03). Mothers that were homozygous for the *s* allele did not exhibit any restraints, indicating a robust effect with each addition of the *s* allele. White bars indicate mothers that were homozygous for the *L* allele, gray bars indicate heterozygous mothers, and black bars indicate mothers that were homozygous for the *s* allele. Error bars are standard errors.

Results indicated a significant effect of maternal *5-HTT* genotype on maternal restraint frequency ($F(2,115) = 4.29$, $p = .02$), with mothers that were homozygous for the *L* allele restraining their infants more, on average ($p < .01$), when compared to heterozygous mothers and mothers homozygous for the *s* allele ($p < .03$). Mothers homozygous for the *s* allele never restrained their infants (*LL*: *M* = 0.16 ± 0.32; *Ls*: *M* = 0.05 ± 0.12; *ss*: *M* = 0.00 ± 0.00; see Fig 2 and S2 Fig).

To determine whether maternal rejections and maternal restraints were separate behavioral dimensions, a bivariate correlation was performed, with results showing that these two behaviors are not significantly correlated ($r = -0.04$, $p = .67$), an indication of independent behavior dimensions, consistent with Fairbanks' review [43].

Results from an overall MANOVA indicated that there was a statistically significant effect of maternal *5-HTT* genotype on maternal rejections and restraints ($F(12,204) = 0.77$, $p = .008$; Wilk's $\Lambda = 0.774$, partial $\eta^2 = .12$).

No significant *5-HTT* genotype effects were found for the other behaviors listed in Table 1.

## Discussion

Overall, the results provide broad support for the hypothesis: prior to an age when weaning typically occurs, when infants are dependent on their mothers for their psychological and

physical needs [49], mothers homozygous for the *s* allele were more likely to reject their infants (see Fig 1) and were less likely to protect their infants from potential harm by restraining them to maintain close proximity (see Fig 2). The infants in this study were very young. Rejections at this age are considered atypical, if not abnormal, and studies show that it leads to aberrant infant behaviors [37]. While sequential behavior coding could not be used, the behavior coders noted that many of the recorded rejections resulted when the infant was seeking their mother for comfort. Most mothers keep such young infants in close proximity to protect them from injury and rough treatment by older female siblings and kin seeking to "*aunt*" the infant, as well as older peers seeking playmates. The results suggest that, at a time when the typical immature rhesus monkey infant is soliciting maternal care and is highly dependent on their mother as a secure base for anxiety reduction, mothers with the *s* allele exhibit more agonistic rejections and less protection of their physically and psychologically needy infants. Given the infants' young age and immature physical and psychological capabilities, such maternal behaviors are in most cases not appropriate and other studies show that it leads to infant psychopathology [37].

As noted in the introduction, results from human studies investigating the effect of the *5-HTT* genotype on mother-infant interactions and attachment are mixed, with some studies [13, 20] showing that mothers with the *s* allele are more likely to engage in insensitive parenting, while others [16, 17] show that mothers with an *s* allele exhibit more maternal sensitivity. These discrepancies have led some to posit that *5-HTT* is a plasticity gene, conveying risk for or protection from negative outcomes, depending on the environmental context [15–17, 24]. The nonhuman primate model offers increased control over social and setting variables, and provides a more homogeneous mother-infant experience than most human studies. To the extent that these results generalize to humans, they suggest that the *s* allele leads to less maternal sensitivity, at least when the mother and infant live in supportive environments surrounded by kin. This interpretation lends support for the Bakermans-Kranenburg and van IJzendoorn's [19], and Morgan, Hammen, and Lee's [20] findings that mothers with one or two copies of the *s* allele are at greater risk for insensitive parenting behaviors, although in these later studies, as in other studies cited earlier in this paper [15–17, 24], this effect is environmentally dependent. Our group has a long history of assessing the interaction between aberrant and normative early environments and the *5-HTT* genotype [30, 50, 51]; however, in the present study all but one of the mothers came from a homogeneous population of females that experienced normative early life rearing conditions. These homogenous early experiences increased the ability to detect small effects, a strength of the study. While the social dominance rank of the mother was not available, it would be of interest to assess the role of maternal social dominance rank as an environmental variable that interacts with *5-HTT* genotype to mediate the role of maternal sensitivity, which may increase the effect size of the *5-HTT* genotype on maternal sensitivity.

As noted earlier, rhesus monkey mothers typically initiate the weaning process after their infants reach three months of age, and high rates of rejections before this age are rare in typical mothers, while high rates of restraints to maintain close intimate contact with their infants is the expected. In the wild, the rate of maternal weaning behaviors peak around the time their infants reach about six months of age, with rejections becoming more frequent, while restraints steadily decline [49, 52]. Studies in rhesus monkeys suggest that infants that are rejected earlier are subject to higher rates of premature mortality, and when they survive, they are more likely to exhibit aberrant behaviors such as aggression, anxiety-like behaviors, and high cortisol [37, 53]. Prematurely rejected infants also tend to show reduced emotional regulation, including high rates of tantrums and high rates of distress when separated from their mothers [41], suggesting that mothers that reject their infants too early are less effective at

providing a secure base from which the infant can learn to effectively regulate arousal and fear. Prior to month three of life, there is likely selective evolutionary pressure on mothers to keep their young infants in close proximity, as those infants that are not kept close are at increased risk for injury, predation, and aggression from older age-mates [43, 49]. Taken together, the findings of this study suggest that, at a time when mothers should provide a secure base to reduce infant fear and anxiety, thus promoting infant exploration and interactions with peers, mothers with the *s* allele may exhibit impaired sensitivity to their infant's needs, as evidenced by increasing rates of maternal rejections and low rates of maternal restraints.

One limitation of the present study is that infant *5-HTT* genotype was only available on a small subset of the infants, precluding the assessment of maternal genotype-by-infant genotype interactions. Some studies in humans suggest that infant genotype also influences maternal sensitivity [20, 54]; however, the relationship between maternal and infant genotypes is complex, and investigations require a large number of subjects, are subject to environmental influences, and effects are difficult to disentangle. Given that the *5-HTT* genotype is passed on from mother-to-infant, some of the infants of the *s* allele mothers also possessed the *s* allele. Thus, infant evocative genotypic effects cannot be ruled out, with mothers modifying their maternal behaviors based on the genetic profile of their offspring. Subsequent studies should obtain genotypes in both mother and infant in a larger sample to assess how maternal genotype interacts with infant genotype to evoke or influence maternal behaviors. While fathers do not demonstrate direct infant care in this species [49], future studies focused on paternal contributions to infant development could assess the effect of fathers' presence in the social group and *5-HTT* genotype as mediators of offspring behavior early in life. The findings of this study are an important step in elucidating the genetic underpinnings of individual differences in maternal sensitivity and highlight the role of maternal *5-HTT* genotype in the quality of infant care.

## Supporting information

**S1 Fig. Individual data points for maternal rejections of infants grouped by maternal *5-HTT* genotype.** Plot depicts jittered individual maternal rejection data points, grouped by maternal *5-HTT* genotype. When compared to mothers that were homozygous for the *L* allele, mothers that were homozygous for the *s* allele exhibited higher rates of infant rejections ($p = .03$). Mothers that were homozygous for the *L* allele also exhibited fewer rejections, when compared to mothers that were heterozygous ($p = .03$). White bars/green data points indicate mothers that were homozygous for the *L* allele, gray bars/blue data points indicate heterozygous mothers, and black bars/orange data points indicate mothers that were homozygous for the *s* allele. Data points are jittered to increase visibility. Error bars are standard errors. (PDF)

**S2 Fig. Individual data points for maternal restraints of infants grouped by maternal *5-HTT* genotype.** Plot depicts jittered individual maternal restraint data points, grouped by maternal 5-HTT genotype. When compared to heterozygous mothers and mothers that were homozygous for the *s* allele, mothers homozygous for the *L* allele exhibited the highest rates of maternal restraints ($p = .02$). Mothers that were homozygous for the *L* allele restraining their infants more, on average ($p < .01$), when compared to heterozygous mothers and mothers homozygous for the *s* allele ($p < .03$). Mothers that were homozygous for the *s* allele did not exhibit any restraints, indicating a robust effect with each addition of the *s* allele. White bars/ green data points indicate mothers that were homozygous for the *L* allele, gray bars/blue data points indicate heterozygous mothers, and black bars/orange data points indicate mothers that were homozygous for the *s* allele. Data points are jittered to increase visibility. Error bars are

standard errors.
(PDF)

**S1 File. Minimal anonymized dataset.**
(XLSX)

## Acknowledgments

We would like to thank the research and animal care staff for their assistance and care of the animals. We would also like to thank the many undergraduates for their contributions to this work, including Alexander Baxter, Elia Hafen, Emily Hepworth, Kelsie Luck, Joseph Reyelts, and Maclean Sherren.

## Author Contributions

**Conceptualization:** Elizabeth K. Wood, J. Dee Higley.

**Data curation:** Elizabeth K. Wood, Ryno Kruger, Colt Halter, Natalia Gabrielle, Laura Del Rosso, John P. Capitanio, J. Dee Higley.

**Formal analysis:** Elizabeth K. Wood, Colt Halter, Natalia Gabrielle, J. Dee Higley.

**Funding acquisition:** John P. Capitanio, J. Dee Higley.

**Methodology:** J. Dee Higley.

**Project administration:** Elizabeth K. Wood, Ryno Kruger, J. Dee Higley.

**Resources:** J. Dee Higley.

**Software:** J. Dee Higley.

**Supervision:** Zachary Baron, J. Dee Higley.

**Visualization:** J. Dee Higley.

**Writing – original draft:** Elizabeth K. Wood, J. Dee Higley.

**Writing – review & editing:** Elizabeth K. Wood, Zachary Baron, Colt Halter, Natalia Gabrielle, Leslie Neville, Ellie Smith, Leah Marett, Miranda Johnson, Laura Del Rosso, John P. Capitanio, J. Dee Higley.

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
