## [Decision Letter · Decision Letter 0]

1 Dec 2022

PONE-D-22-27260Variation in the serotonin transporter genotype is associated with maternal restraint and rejection of infants: A nonhuman primate (Macaca mulatta)modelPLOS ONE

Dear Dr. Wood,

Thank you for submitting your manuscript to PLOS ONE. After careful consideration, we feel that it has merit but does not fully meet PLOS ONE’s publication criteria as it currently stands. Therefore, we invite you to submit a revised version of the manuscript that addresses the points raised during the review process. When revising your manuscript, please consider all issues mentioned in the reviewers' comments carefully: please outline every change made in response to their comments and provide suitable rebuttals for any comments not addressed. Please note that your revised submission may need to be re-reviewed. 

PLOS ONE values your contribution, and we look forward to receiving your revised manuscript.

We look forward to receiving your revised manuscript.

Kind regards,

Tamas Kozicz

Academic Editor

PLOS ONE

Journal Requirements:

“This work was supported by: R24OD010962 (JPC) and P51OD011157 (CNPRC base grant), as well as small mentoring grants from Brigham Young University.”

 “This work was supported by: R24OD010962 (JPC) and P51OD011157 (CNPRC base grant), as well as small mentoring grants from Brigham Young University. The funders had no role in study design, data collection and analysis, decision to publish, or preparation of the manuscript.”

Reviewers' comments:

Reviewer's Responses to Questions

**Comments to the Author**

1. Is the manuscript technically sound, and do the data support the conclusions?

Reviewer #1: Yes

Reviewer #2: Partly

2. Has the statistical analysis been performed appropriately and rigorously? 

Reviewer #1: Yes

Reviewer #2: Yes

3. Have the authors made all data underlying the findings in their manuscript fully available?

Reviewer #1: Yes

Reviewer #2: Yes

4. Is the manuscript presented in an intelligible fashion and written in standard English?

Reviewer #1: Yes

Reviewer #2: Yes

5. Review Comments to the Author

Reviewer #1: The authors here study the influence of 5-HTT genotype variation on maternal sensitivity in parenting in a nonhuman primate model (Rhesus monkey) during the first three months of their infant’s lives. The authors hypothesized that, at an age when infants are still dependent on their mother to act as a secure base and to protect them, mothers possessing the s allele of the 5-HTT genotype will exhibit inappropriate parenting behaviours, such as premature maternal rejections of their immature, dependent infants. As a corollary, it is hypothesized that rhesus mothers with the s allele will be less likely to keep their infant in proximity, as measured by restraint and retrieving their infants’ attempts at exploration.

Results showed that mothers homozygous for the s allele rejected their infants the most and restrained their infants the least, an indication that mothers with the s allele are more likely to neglect their infants’ psychological and physical needs.

One of the strengths of the study is the number of individuals (N = 125 dyad mother – infant);

n = 50 mothers were homozygous for the L allele, n = 58 mothers were heterozygotes, and n =14 mothers were homozygous for the s allele. Three subjects possessed rare 5-HTT variants (i.e., XL, XXL), and they were excluded from the analyses.

This is a well writing and design research. The findings of this research add to the comprehension of the role of the genetic underpinnings individual differences in maternal sensitivity and suggest that maternal 5-HTT genotype plays a role in the quality of infant care. I only have minor comments.

• It would be more informative to present the figures (graph) by showing both individual points and bars (to see the difference between individual).

• By curiosity, in this primate model does the father also play a role in infant care?

Reviewer #2: This study aims to understand the impact of 5-HTT genotype on maternal behaviors. The authors report that mothers homozygous for the s allele rejected their infants the most and restrained their infants the least, indications that mothers with the s allele are more likely to neglect their infants’ psychological and physical needs. The results are significant and advance our understanding of the etiology of variability in maternal warmth and care. In addition, the results suggests that maternal 5-HTT genotype should be examined in studies assessing genetic influences on variation in maternal sensitivity, and ultimately, mother-infant attachment quality

Major:

As the authors state in the discussion, results from human studies investigating the effect of the 5-HTT genotype on mother-infant interactions and attachment are mixed, with studies showing that mothers with the s allele are more likely to engage in insensitive parenting, while other studies show that mothers with an s allele exhibit more maternal sensitivity. In rodents it has also been shown that 5-HTT s-allele carriers have an increased risk to develop depression when exposed to ELS suggesting that carrying the s-allele does not inevitably have negative consequences. Rather, the increased sensitivity of s-allele carriers to a respective match or mismatch between the early and adult life environment may govern their adaptive or maladaptive responses to stress (PMID: 23319004). Overall, these discrepancies have led some to posit that 5-HTT as a plasticity gene, conveys risk for or protection from negative outcomes, depending on the environmental contexts. Therefore, the authors need to report on or control for maternal environmental stress and early life adversity in their analysis. Controlling for these factors may nuance their findings and further help us to resolve the issue whether 5-HTT as a plasticity gene governs adaptive or maladaptive maternal behaviors.

6. PLOS authors have the option to publish the peer review history of their article (what does this mean?). If published, this will include your full peer review and any attached files.

Reviewer #1: No

Reviewer #2: **Yes: **Tamas Kozicz

---

## [Author Response · Author response to Decision Letter 0]

12 Jan 2023

Manuscript PONE-D-22-27260

Response to Reviewers

January 10, 2023

Dear Dr. Kozicz, 

Thank you for giving us an opportunity to submit a revised draft of the manuscript, “Variation in the serotonin transporter genotype is associated with maternal restraint and rejection of infants: A nonhuman primate (Macaca mulatta) model” to PLOS ONE. We appreciate the time and effort that you and the reviewers dedicated to providing feedback on our manuscript. We have reviewed and incorporated reviewers’ suggestions. Those changes are highlighted within the manuscript. Please see below, in blue, a point-by-point response to the journal requirements and the reviewers’ comments and concerns. All page numbers refer to the revised manuscript file with the highlighted changes. 

Thank you, 

Dr. Elizabeth Wood

Journal Requirements

Thank you for pointing this out. We have reviewed PLOS ONE’s style requirements and adjusted the manuscript accordingly. The modifications are highlighted in the revised version of the manuscript. 

2. Thank you for stating the following in the Acknowledgments Section of your manuscript: “This work was supported by: R24OD010962 (JPC) and P51OD011157 (CNPRC base grant), as well as small mentoring grants from Brigham Young University.” We note that you have provided additional information within the Acknowledgements Section that is not currently declared in your Funding Statement. Please note that funding information should not appear in the Acknowledgments section or other areas of your manuscript. We will only publish funding information present in the Funding Statement section of the online submission form. Please remove any funding-related text from the manuscript and let us know how you would like to update your Funding Statement. Currently, your Funding Statement reads as follows: “This work was supported by: R24OD010962 (JPC) and P51OD011157 (CNPRC base grant), as well as small mentoring grants from Brigham Young University. The funders had no role in study design, data collection and analysis, decision to publish, or preparation of the manuscript.” Please include your amended statements within your cover letter; we will change the online submission form on your behalf.

We apologize for this error. We have removed the statement from the acknowledgements section of the manuscript. The funding statement should continue to read, “This work was supported by: R24OD010962 (JPC) and P51OD011157 (CNPRC base grant), as well as small mentoring grants from Brigham Young University. The funders had no role in study design, data collection and analysis, decision to publish, or preparation of the manuscript.”

To address this comment, we now provide a minimal anonymized dataset as a supporting information file (S1 File). 

Reviewers' comments:

1. Reviewer #1: The authors here study the influence of 5-HTT genotype variation on maternal sensitivity in parenting in a nonhuman primate model (Rhesus monkey) during the first three months of their infant’s lives. The authors hypothesized that, at an age when infants are still dependent on their mother to act as a secure base and to protect them, mothers possessing the s allele of the 5-HTT genotype will exhibit inappropriate parenting behaviours, such as premature maternal rejections of their immature, dependent infants. As a corollary, it is hypothesized that rhesus mothers with the s allele will be less likely to keep their infant in proximity, as measured by restraint and retrieving their infants’ attempts at exploration. Results showed that mothers homozygous for the s allele rejected their infants the most and restrained their infants the least, an indication that mothers with the s allele are more likely to neglect their infants’ psychological and physical needs.

One of the strengths of the study is the number of individuals (N = 125 dyad mother – infant); n = 50 mothers were homozygous for the L allele, n = 58 mothers were heterozygotes, and n =14 mothers were homozygous for the s allele. Three subjects possessed rare 5-HTT variants (i.e., XL, XXL), and they were excluded from the analyses. This is a well writing and design research. The findings of this research add to the comprehension of the role of the genetic underpinnings individual differences in maternal sensitivity and suggest that maternal 5-HTT genotype plays a role in the quality of infant care. 

I only have minor comments.

It would be more informative to present the figures (graph) by showing both individual points and bars (to see the difference between individual).

We now provide individual data points in bar graphs in order to improve data visualization and increase transparency. These modifications are depicted in the supporting information of the manuscript as follows: 

By curiosity, in this primate model does the father also play a role in infant care?

In this species, mothers are the primary caregivers, while fathers have a peripheral role. We have updated the discussion section of the manuscript to reflect this curiosity by including the following statement, “While fathers do not demonstrate direct infant care in this species (Lindburg, 1971), future studies focused on paternal contributions to infant development could assess the effect of fathers’ presence in the social group and 5-HTT genotype as mediators of offspring behavior early in life.”

2. Reviewer #2: This study aims to understand the impact of 5-HTT genotype on maternal behaviors. The authors report that mothers homozygous for the s allele rejected their infants the most and restrained their infants the least, indications that mothers with the s allele are more likely to neglect their infants’ psychological and physical needs. The results are significant and advance our understanding of the etiology of variability in maternal warmth and care. In addition, the results suggests that maternal 5-HTT genotype should be examined in studies assessing genetic influences on variation in maternal sensitivity, and ultimately, mother-infant attachment quality

As the authors state in the discussion, results from human studies investigating the effect of the 5-HTT genotype on mother-infant interactions and attachment are mixed, with studies showing that mothers with the s allele are more likely to engage in insensitive parenting, while other studies show that mothers with an s allele exhibit more maternal sensitivity. In rodents it has also been shown that 5-HTT s-allele carriers have an increased risk to develop depression when exposed to ELS suggesting that carrying the s-allele does not inevitably have negative consequences. Rather, the increased sensitivity of s-allele carriers to a respective match or mismatch between the early and adult life environment may govern their adaptive or maladaptive responses to stress (PMID: 23319004). Overall, these discrepancies have led some to posit that 5-HTT as a plasticity gene, conveys risk for or protection from negative outcomes, depending on the environmental contexts. Therefore, the authors need to report on or control for maternal environmental stress and early life adversity in their analysis. Controlling for these factors may nuance their findings and further help us to resolve the issue whether 5-HTT as a plasticity gene governs adaptive or maladaptive maternal behaviors.

We appreciate this comment and agree that early life conditions and 5-HTT genotypic effects are important to consider in tandem. To address this, in the Methods section we note the following, “With the exception of one subject that was reared in the nursery before introduction into the larger, outdoor social group into which her infant was born, the early life history of all of the subjects was identical, including being born into and reared in the large, outdoor corrals”. As well as, “As noted above, while the majority of mothers were born and reared as infants in the larger corrals, one mother (heterozygous for the 5-HTT genotype) was reared in the nursery. Preliminary analyses assessing the impact of including this subject showed no difference in the direction or significance of effects, so they were included in final models”. 

In the discussion, we also added, ”Our group has a long history of assessing the interaction between aberrant and normative early environments and the 5-HTT genotype (Bennett et al, 2002; Hunter et al, 2022; Wood et al 2021); however, in the present study all but one of the mothers came from a homogeneous population of females that experienced normative early life rearing conditions. These homogenous early experiences increased our ability to detect small effects, a strength of the study.”

---

## [Decision Letter · Decision Letter 1]

5 Feb 2023

Variation in the serotonin transporter genotype is associated with maternal restraint and rejection of infants: A nonhuman primate (*Macaca mulatta*) model

PONE-D-22-27260R1

Dear Dr. Wood,

We’re pleased to inform you that your manuscript has been judged scientifically suitable for publication and will be formally accepted for publication once it meets all outstanding technical requirements.

Kind regards,

Tamas Kozicz

Academic Editor

PLOS ONE

Additional Editor Comments (optional):

Reviewers' comments:

Reviewer's Responses to Questions

**Comments to the Author**

1. If the authors have adequately addressed your comments raised in a previous round of review and you feel that this manuscript is now acceptable for publication, you may indicate that here to bypass the “Comments to the Author” section, enter your conflict of interest statement in the “Confidential to Editor” section, and submit your "Accept" recommendation.

Reviewer #1: All comments have been addressed

Reviewer #2: All comments have been addressed

2. Is the manuscript technically sound, and do the data support the conclusions?

Reviewer #1: Yes

Reviewer #2: Yes

3. Has the statistical analysis been performed appropriately and rigorously? 

Reviewer #1: Yes

Reviewer #2: Yes

4. Have the authors made all data underlying the findings in their manuscript fully available?

Reviewer #1: Yes

Reviewer #2: Yes

5. Is the manuscript presented in an intelligible fashion and written in standard English?

Reviewer #1: Yes

Reviewer #2: Yes

6. Review Comments to the Author

Reviewer #1: In this revised version of the MS all the concerns and point raised were addressed which have improved the manuscript.

Reviewer #2: (No Response)

7. PLOS authors have the option to publish the peer review history of their article (what does this mean?). If published, this will include your full peer review and any attached files.

Reviewer #1: No

Reviewer #2: **Yes: **Tamas Kozicz M.D., Ph.D.

---

## [Editor Report · Acceptance letter]

14 Apr 2023

PONE-D-22-27260R1 

Variation in the serotonin transporter genotype is associated with maternal restraint and rejection of infants: A nonhuman primate (*Macaca mulatta*) model 

Dear Dr. Wood:

I'm pleased to inform you that your manuscript has been deemed suitable for publication in PLOS ONE. Congratulations! Your manuscript is now with our production department. 

Kind regards, 

on behalf of

Dr. Tamas Kozicz 

Academic Editor

PLOS ONE